# Optimizing digital implant impressions: Evaluating the significance of scan body image deficiency and alignment under varied scan body exposures

**Pobploy Petchmedyai, Prakan Thanasrisuebwong**[ORCID]*

Dental Implant Centre, Faculty of Dentistry, Mahidol University, Salaya, Thailand

* prakan.tha@mahidol.ac.th

**Data Availability Statement:** We have uploaded the raw data set of 3-D measurements collected from the 3-D analysis software as a supporting

## Abstract

In implant dentistry, the advent of intraoral scanning technology has revolutionized traditional clinical processes by streamlining procedures and ensuring predictable treatment outcomes. However, achieving accurate virtual implant positions using intraoral scanners and scan bodies can be influenced by various clinical and laboratory factors. This study aims to investigate the impact of scan body image capture deficiency and scan body alignment methods in computer-aided design (CAD) software on the accuracy of virtual implant positions, particularly in different implant depths. Three stereolithographic half-arch implant models with different implant depths were prepared, representing three scenarios of scan body exposure: full exposed scan body, 2/3 exposed scan body, and 1/3 exposed scan body. The scan body image capture deficiency and alignment methods were simulated using CAD software. The deviation of virtual implant positions obtained from different scenarios were evaluated using 3D analysis software. The highest angular and linear deviation (0.237±0.059 degrees, 0.084±0.068 mm) were found in the 1/4 upper and lower part scan body deficiency using the 1-point alignment method in the 1/3 exposed scan body. Two-way ANOVA analysis revealed significant effects of scan deficiency on virtual implant position deviations across all scan body exposures, except for the linear deviation when the scan body was exposed 2/3 of its length. Furthermore, scan deficiencies in the 1/4 upper and lower parts of the scan body significantly affected implant angular deviation regardless of scan body exposure, while implant linear deviation was specifically affected when the scan body was exposed to only 1/3 of its total length. Deficiencies in scan body acquisition, particularly in deep soft tissue situations, can lead to deviations in both angular and linear positioning of virtual implants. Employing appropriate scan body alignment methods such as a 3-point alignment approach demonstrates better accuracy compared to a 1-point alignment.

## Introduction

Advancements in computer-aided design and computer-aided manufacturing (CAD/CAM) technology have revolutionized dental workflows with the introduction of intraoral scanners for digital impressions. These scanners capture precise three-dimensional (3D) information of

document, in compliance with PLOS ONE's data policy. This will enable transparency and further analysis by interested researchers.

**Funding:** Funded by the Faculty of Dentistry, Mahidol University. The funders had no role in study design, data collection and analysis, decision to publish, or preparation of the manuscript.

**Competing interests:** The authors have declared that no competing interests exist.

intraoral structures, eliminating the need for traditional laboratory processes and enhancing efficiency. The digital files can be stored electronically or used for fabricating physical models using additive or subtractive manufacturing methods [1]. This technology not only increases productivity and reduces material waste but also improves patient comfort and acceptance [2, 3]. In implant dentistry, digital technology is employed to capture intraoral implant positions for prostheses fabrication. However, directly scanning the implant itself is not feasible due to its location below the gingival level. To overcome this, specific scanning components called "scan bodies" are utilized to represent the implant's 3D position and orientation. These scan bodies, commonly provided by implant manufacturers, have a predefined geometry and dimension that allows their CAD models to be imported into dental CAD software libraries. When the scanned data is imported, it can be aligned with the scan body CAD model to correct any scanning errors. Subsequently, the scan body is digitally subtracted, and the implant is positioned based on the scan body's position. This approach ensures accurate implant position recording and facilitates efficient prostheses fabrication using CAD/CAM technology [4, 5].

The accuracy of virtual implant positions obtained from intraoral scanners is crucial for ensuring proper prosthesis fit and the long-term success of implant prostheses [6, 7]. When capturing implant position using intraoral scanner, clinicians commonly employ an implant scan body provided by the implant manufacturer, typically made of polymer materials like polyether ether ketone (PEEK) [5]. In an in vitro study specifically focused on single unit implants, the average accuracy of transferring the scanned position to a 3D printed implant cast using PEEK scan bodies was observed to range from approximately 105–127 μm, with an angular deviation of 0.22˚-1.25˚ [8]. In vitro studies have demonstrated that digital implant impressions using intraoral scanners exhibit comparable accuracy to conventional impressions, provided that the achieved accuracy remains below 150 μm [9–11]. However, ongoing research is investigating the factors that influence scanning accuracy, particularly in various clinical scenarios. Challenges arise when using intraoral scanners in the oral cavity, especially when scanning malpositioned implants, as limited access can make it difficult to capture the entire surface of the scan body. Moreover, the size of the scanning head of an intraoral scanner can present difficulties. Some scanners have relatively larger scanning heads, which can make it difficult to access and capture the complete surface of the scan body, especially in areas with restricted access especially Molar area. In addition to the scanning head size, patient-related factors can also affect the completeness of the scan. Limited mouth opening is one such factor. Patients with conditions such as temporomandibular joint (TMJ) disorders, muscle tightness, or previous surgical procedures may experience reduced jaw mobility. As a result, it becomes challenging to properly position the intraoral scanner. In such scenarios, it has been observed that the virtual implant deviation could potentially reach up to 80 microns, with the scan body image deficiency being identified in only 15% of total scan body surface [12, 13]. Implant depth adds another layer of complexity to the scanning process. In specific clinical cases, such as the anterior esthetic zone or areas with limited bone availability, implants may need to be placed at greater depths. This can result in reduced scan body exposure during image capture with intraoral scanners. However, the influence of implant depth on the accuracy of virtual implant positioning remains inconclusive, with contradictory findings reported in various studies [4, 14–16]. In the study by Giménez et al. [14], utilizing parallel confocal laser technology with the iTero scanner, the mean distance error for the 0-mm implant depth was 23.1 μm, for the 2-mm depth it was 16.2 μm, and for the 4-mm depth it was 27.9 μm. This study found that the 0-mm depth exhibited lower accuracy compared to the 2-mm depth, and accuracy decreased with longer scans, potentially due to inaccuracy resulting from the stitching processes. In another study by Giménez et al. [16], employing active wavefront sampling technology with the Lava Cos scanner, the mean distance error for the 0-mm depth was 28.3 μm, and

for the 2-mm depth it was 34.3 μm. This study found that the 0-mm depth was more accurate than the 2-mm.

This study aims to analyze the impact of scan body image capture deficiency and scan body alignment methods in CAD software on the accuracy of digital implant impressions, with a focus on different implant depths. By comprehensively assessing these factors, the study aims to identify potential sources of error in the digital implant workflow and develop an optimal scanning strategy to minimize these errors. The hypothesis was that there was no significant difference in the accuracy of digital implant impressions based on scan body image capture deficiency and scan body alignment methods, irrespective of different implant depths. The findings of this research will contribute to improving the accuracy and reliability of digital implant impressions, leading to enhanced treatment outcomes in implant dentistry.

## Materials and methods

### Study workflow and master cast fabrication

The detailed workflow of this study was represented in Fig 1.

Stereolithographic half-arch implant casts of the second mandibular premolar were prepared as master casts with different implant depths to represent three scenarios of scan body exposure: full exposed scan body (exposed scan body length = 8mm), 2/3 exposed scan body (exposed scan body length = 5.33mm), and 1/3 exposed scan body (exposed scan body length = 2.67mm) (Fig 2). In order to control the depth of implant placement and create consistent models for the different scan body exposure scenarios, CAD design software was utilized. The implant depths were precisely adjusted in the software to ensure accurate representation of the desired scan body exposure lengths.

### Cast digitization using a reference laboratory scanner and an intraoral scanner

The casts with scan bodies were digitized using a laboratory scanner with 4 μm accuracy (Ceramill Map 600, Amann Girrbach AG, Austria) to obtain reference scan data for each master implant model. The intraoral scanner (TRIOS® 3; 3 Shape, Denmark) was used to scan the master implant models ten times for each model to be as tested scanned data. All scanned data were exported in stereolithography (STL) file format.

### Generating scan body image deficiency

In this study, the researchers aimed to simulate scan body deficiencies by intentionally creating these deficiencies in CAD software to replicate a clinical situation where the intraoral scanner's size or positioning limitations prevent the accurate capture of the entire scan body geometry. To create different scenarios for varying scan body deficiency, STL files were generated using an intraoral scanner and processed in CAD software (Meshmixer 2017 version 3.5.474, Autodesk, USA). Two modifications were made to the scan body surfaces: 1) removing 1/4 of the upper and lower parts, and 2) removing 1/4 of the lower part while retaining the flat surface of the scan body. This resulted in a total of 90 scanned data samples obtained from nine different simulated implant models, as illustrated in Fig 3.

To ensure reproducibility, a standardized method in the CAD software to create scan body deficiencies was followed. The method involved utilizing the "plane cutting" technique to remove specific portions of the scan body, aligning the cut plane with the cylinder's center axis, the "Snap to Grid" was employed to improve alignment accuracy with reference points and saving the modified STL files for subsequent 3D measurements.

Implant model fabriation (Figure2)

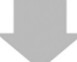

Model digitization using a reference laboratory scanner and an intraoral scanner

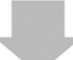

Generating scan body image deficiency in 9 different scenarios (Figure3)

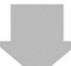

Converting scan body position to implant position using two scan body alignment methods (Figure 4)

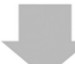

Creating virtual implant model (Figure5)

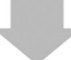

Accuracy assessment of virtual implant positions in terms of angular deviation and linear deviation (Figure 6)

**Fig 1. Workflow of the study.**

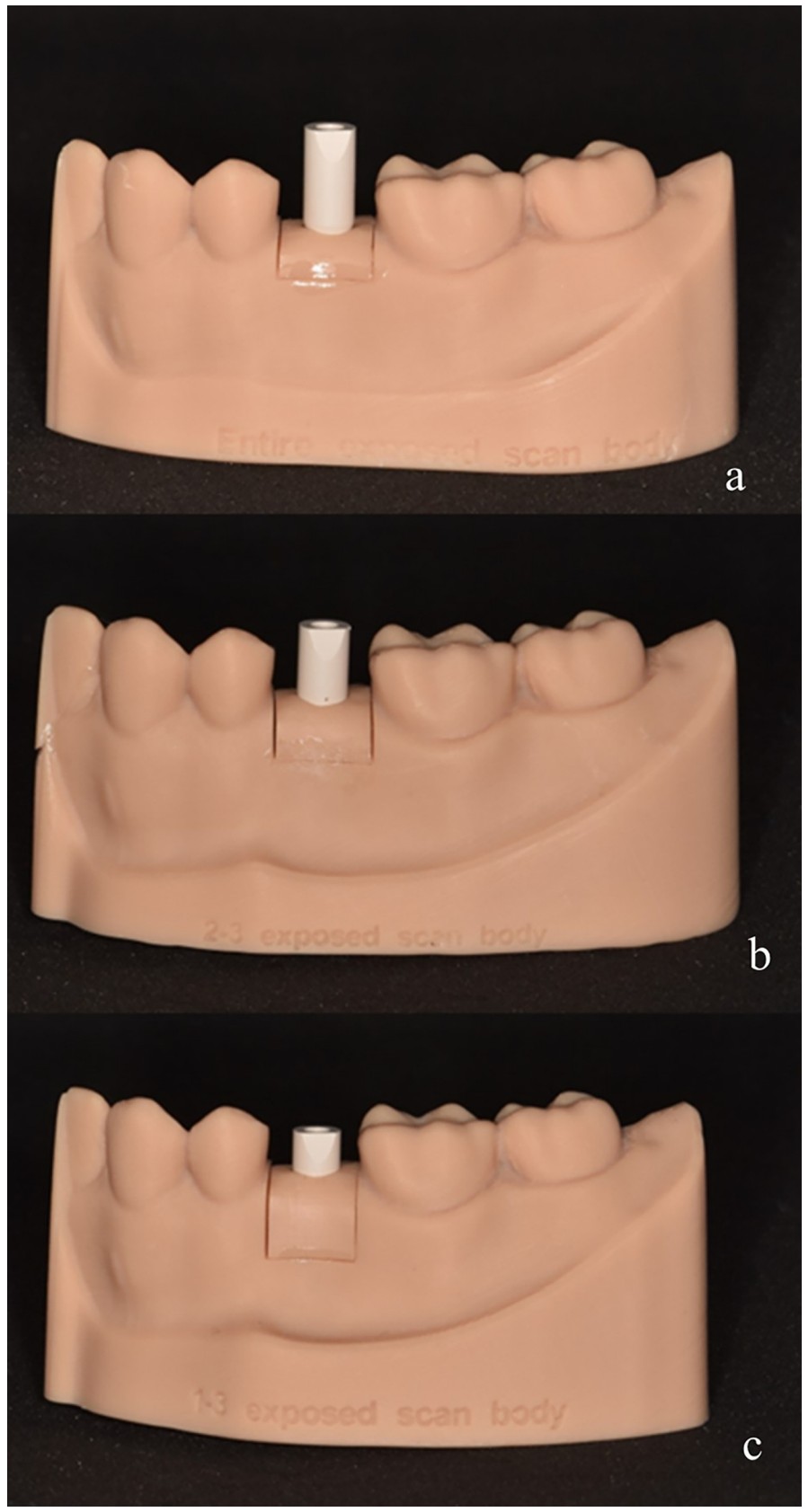

**Fig 2.** Three implant models with different scan body exposures: (a) full exposed scan body, (b) 2/3 exposed scan body, (c) 1/3 exposed scan body.

## Creating virtual implant models

The scanned data of each scenario were imported into dental CAD software (3 Shape dental system® 2019; 3 Shape, Denmark) to create the virtual implant model, the scan body positions were converted into implant positions. Two different scan body alignment methods available in the software; 1) 1-point alignment and 2) 3-point alignment, as shown in Fig 4, were used to align the scan body to the reference scan body in the scan body library of the software. A total of 180 virtual implant models (Fig 5) were obtained.

## Accuracy assessment of virtual implant positions

All virtual implant models were exported in STL file format and subsequently imported into the 3-D analysis software (GOM inspect, version 2018; GOM, Germany) for data analysis with one operator to keep consistency in the measurements. The superimposition of reference scanned data and tested scanned data were performed using the pre-alignment function. An angular deviation of implant position was measured between two cylindrical axes of the virtual implants. The linear deviation of implant position was measured between two intersection

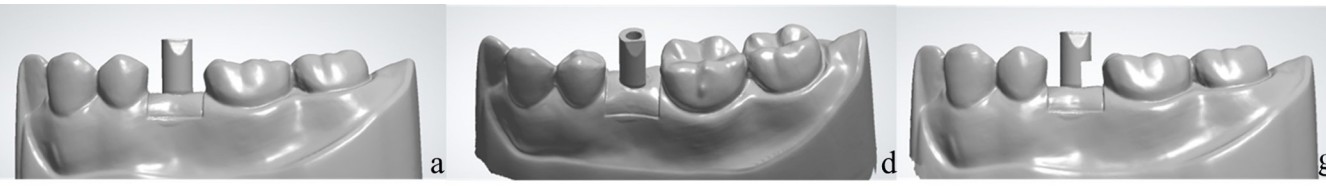

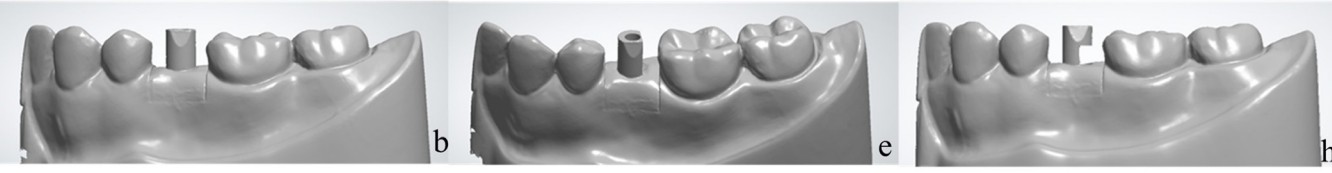

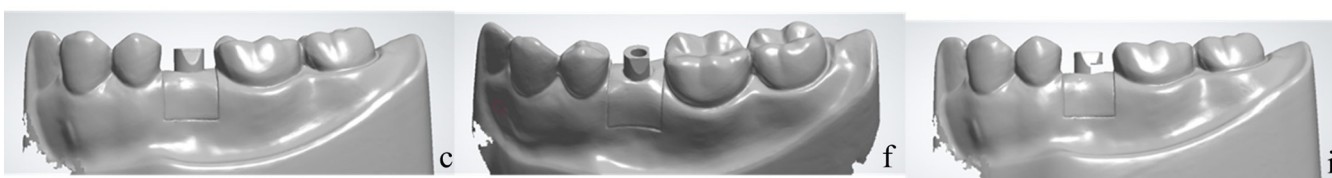

**Fig 3. Nine different scenarios of scan body exposure and deficiency were created in CAD software.** The virtual scan body image deficiencies were intentionally generated to reflect clinical situations where the complete capture of the scan body is not possible. The implant position was varied at three different depths within the model.; (a) full exposed scan body with no scan body deficiency (b) 2/3 exposed scan body with no scan body deficiency (c) 1/3 exposed scan body with no scan body deficiency (d) full exposed scan body with 1/4 upper and lower part of scan body deficiency (e) 2/3 exposed scan body with 1/4 upper and lower part of scan body deficiency (f) 1/3 exposed scan body with 1/4 upper and lower part of scan body deficiency (g) full exposed scan body with 1/4 lower part of scan body deficiency (h) 2/3 exposed scan body with 1/4 lower part of scan body deficiency (i) 1/3 exposed scan body with 1/4 lower part of scan body deficiency.

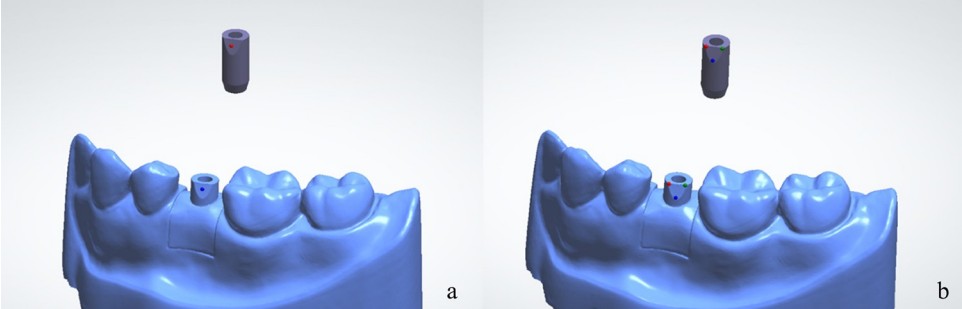

**Fig 4.** Two different scan body alignment methods were used in this study: (a) 1-point alignment and (b) 3-point alignment.

points constructed from a cylindrical axis and a top plane of the implants. The means and standard deviations (SD) of angular and linear deviations of the scan body image were calculated and reported to reflect the accuracy of virtual implant positions obtained from different scenarios. The measurement was demonstrated in Fig 6.

In this study, all scans were performed by a single experienced operator who had sufficient expertise in using intraoral scanners and CAD software. The operator has undergone training and had prior experience in conducting scans, using CAD software and measurements in 3D analysis software.

## Statistical analysis

Statistical analysis was conducted using IBM® SPSS® Statistics version 25 (USA). To assess the accuracy of virtual implant positions, a two-way analysis of variance (ANOVA) and Post hoc analysis using the Tukey Honestly Significant Difference (HSD) was performed. The significance level was set at 0.05. The analysis aimed to identify any differences in accuracy based on deficiencies in scan body image capture and scan body alignment methods in the three levels of scan body exposures.

## Results

The linear and angular deviations of the implants in the different scan body exposures and scan body image capture deficiency when applying two methods of scan body alignment are

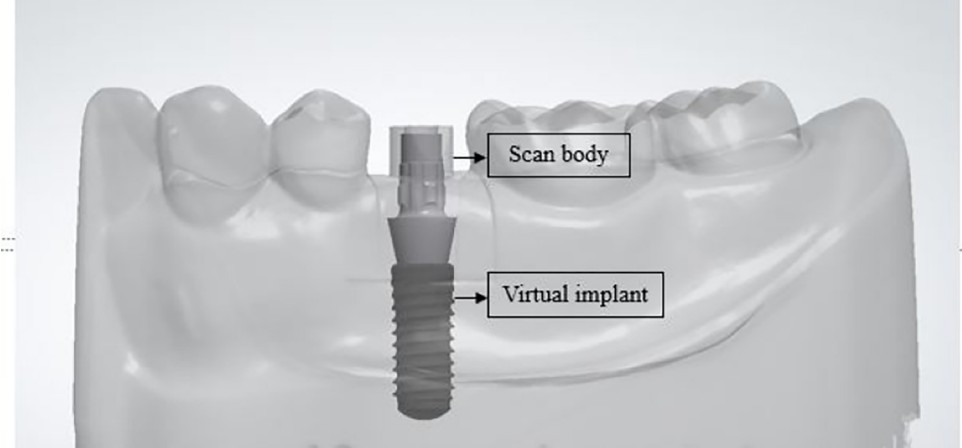

**Fig 5. Virtual implant model obtained after scan body alignment process.**

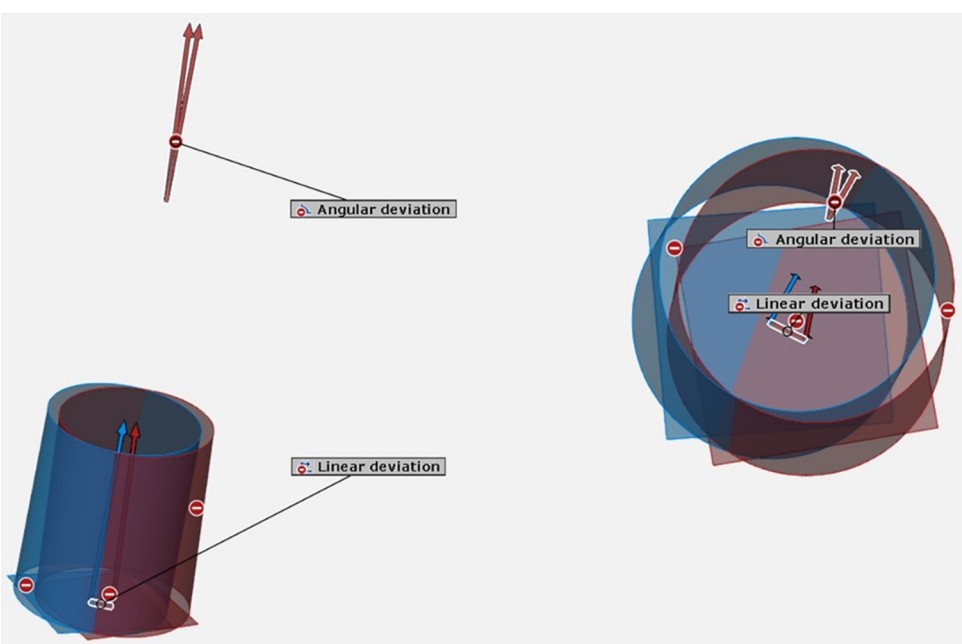

**Fig 6. Angular and linear deviations of virtual implant position measurements.**

shown in Table 1. For angular deviation, the range of values varied depending on scan body exposure and deficiency. The lowest angular deviation was observed in the 2/3 exposed group with ¼ lower part of scan body deficiency using the 1-point alignment method (0.062±0.012 degrees). Conversely, the highest angular deviation was found in the 1/4 upper and lower part scan body deficiency using the 1-point alignment method in 1/3 scan body exposed (0.237

**Table 1. Mean ± SD of angular (degree) and linear (millimeter) implant deviation across different levels of scan body exposure, deficiencies, and using either 3-point or 1-point alignment methods in CAD software.**

| Implant Deviation | Scan body exposure | Scan body deficiency | 3-point alignment | 1-point alignment |
|---|---|---|---|---|
| Angulation (degree) | 1/3 exposed | No deficiency | 0.121±0.037 | 0.122±0.041 |
| | | ¼ lower part | 0.087±0.039 | 0.201±0.107 |
| | | ¼ upper and lower part | 0.162±0.040 | 0.237±0.059 |
| | 2/3 exposed | No deficiency | 0.103±0.009 | 0.101±0.013 |
| | | ¼ lower part | 0.063±0.012 | 0.062±0.012 |
| | | ¼ upper and lower part | 0.130±0.021 | 0.143±0.052 |
| | full exposed | No deficiency | 0.157±0.020 | 0.158±0.023 |
| | | ¼ lower part | 0.116±0.023 | 0.115±0.021 |
| | | ¼ upper and lower part | 0.177±0.040 | 0.181±0.036 |
| Linear (millimeter) | 1/3 exposed | No deficiency | 0.020±0.006 | 0.020±0.006 |
| | | ¼ lower part | 0.012±0.005 | 0.037±0.019 |
| | | ¼ upper and lower part | 0.013±0.005 | 0.084±0.068 |
| | 2/3 exposed | No deficiency | 0.026±0.003 | 0.028±0.005 |
| | | ¼ lower part | 0.025±0.004 | 0.026±0.004 |
| | | ¼ upper and lower part | 0.030±0.007 | 0.030±0.007 |
| | full exposed | No deficiency | 0.030±0.005 | 0.030±0.005 |
| | | ¼ lower part | 0.030±0.004 | 0.030±0.004 |
| | | ¼ upper and lower part | 0.034±0.004 | 0.033±0.003 |

**Table 2. Effects of scan deficiency and scan body alignment methods on the implant deviations calculated by two-way ANOVA model; (Asterisk(*): *P* <0.05).**

| Implant deviation | Scan body exposure | Effect | Df | F-Ratio | P Value |
|---|---|---|---|---|---|
| Angulation (degree) | 1/3 exposed | scan deficiency | 2 | 9.146 | <0.001* |
| | | alignment | 1 | 17.071 | <0.001* |
| | | scan deficiency * alignment | 2 | 4.674 | 0.013* |
| | 2/3 exposed | scan deficiency | 2 | 44.494 | <0.001* |
| | | alignment | 1 | 0.27 | 0.605 |
| | | scan deficiency * alignment | 2 | 0.571 | 0.569 |
| | full exposed | scan deficiency | 2 | 26.345 | <0.001* |
| | | alignment | 1 | 0.034 | 0.855 |
| | | scan deficiency * alignment | 2 | 0.040 | 0.961 |
| Linear (millimetre) | 1/3 exposed | scan deficiency | 2 | 5.560 | 0.006* |
| | | alignment | 1 | 18.267 | <0.001* |
| | | scan deficiency * alignment | 2 | 7.595 | 0.001* |
| | 2/3 exposed | scan deficiency | 2 | 3.143 | 0.051 |
| | | alignment | 1 | 0.283 | 0.597 |
| | | scan deficiency * alignment | 2 | 0.185 | 0.832 |
| | full exposed | scan deficiency | 2 | 4.725 | 0.013* |
| | | alignment | 1 | 0.035 | 0.852 |
| | | scan deficiency * alignment | 2 | 0.012 | 0.988 |

±0.059 degrees). Regarding linear implant deviation, the range of values also differed based on scan body exposure, deficiency, and alignment method. The lowest linear deviation was observed in the 1/3 exposed scan body with 1/4 lower part deficiency using 3-point alignment method (0.012±0.005 mm). On the other hand, the highest linear deviation was found in the 1/3 exposed scan body with 1/4 upper and lower part of scan body deficiency using 1-point alignment method (0.084±0.068mm).

Two-way ANOVA analysis in Table 2. indicated that the effect of the scan deficiency was significant for virtual implant position deviations in all levels of scan body exposures except the linear deviation of implant position when the scan body was exposed 2/3 of the entire length. When utilizing the 1-point scan body alignment technique to position the implant replica, the deviations in the virtual implant were considerably higher in comparison to the 3-point alignment method, specifically when the scan body was exposed to only 1/3 of its complete extent. Additionally, when limiting the scan body exposure to a third of its overall length, there is a noticeable pattern whereby a greater degree of scan deficiency correlates with increased occurrences of implant deviation when 1- point alignment was utilized.

It can be observed from Post hoc analysis by Tukey HSD in Table 3. that when there is a scan deficiency in the 1/4 upper and lower part of the scan body, implant angular deviation appears to be affected regardless of the scan body exposure while implant linear deviation appears to be affected specifically when the scan body is exposed to only 1/3 of the total length.

## Discussion

The main objective of this study was to evaluate the impact of soft tissue depth on the accuracy of implant scan body digital impressions. The findings indicate that when there is deficient acquisition of the scan body due to deep soft tissue depth, it can result in deviations in both angular and linear positioning of virtual implants. However, when the scan body is exposed 2/3 of its length, the statistical analysis yielded a *P* value of 0.051, which is slightly above the predetermined significance threshold of 0.05. Although this result does not reach statistical

**Table 3. Post hoc analysis was performed using the Tukey HSD test to determine significant differences of implant deviations among three levels of scan deficiency in three levels scan body exposure; (Asterisk(*): $P < 0.05$).**

| Implant deviation | Scan body exposure | Scan deficiency | Scan deficiency | P value | 95% Confidence Interval | |
|---|---|---|---|---|---|---|
| | | | | | Lower Bound | Upper Bound |
| Angulation (degree) | 1/3 exposed | No deficiency | ¼ lower part | 0.459 | -0.068 | 0.023 |
| | | | ¼ upper and lower part | <0.001* | -0.123 | -0.033 |
| | | ¼ lower part | No deficiency | 0.459 | -0.023 | 0.068 |
| | | | ¼ upper and lower part | 0.013* | -0.101 | -0.010 |
| | | ¼ upper and lower part | No deficiency | <0.001* | 0.033 | 0.123 |
| | | | ¼ lower part | 0.013* | 0.010 | 0.101 |
| | 2/3 exposed | No deficiency | ¼ lower part | <0.001* | 0.021 | 0.058 |
| | | | ¼ upper and lower part | <0.001* | -0.053 | -0.016 |
| | | ¼ lower part | No deficiency | <0.001* | -0.058 | -0.021 |
| | | | ¼ upper and lower part | <0.001* | -0.093 | -0.055 |
| | | ¼ upper and lower part | No deficiency | <0.001* | 0.016 | 0.053 |
| | | | ¼ lower part | <0.001* | 0.055 | 0.093 |
| | full exposed | No deficiency | ¼ lower part | <0.001* | 0.021 | 0.063 |
| | | | ¼ upper and lower part | 0.049* | -0.043 | 0.000 |
| | | ¼ lower part | No deficiency | <0.001* | -0.063 | -0.021 |
| | | | ¼ upper and lower part | <0.001* | -0.085 | -0.042 |
| | | ¼ upper and lower part | No deficiency | 0.049* | 0.000 | 0.043 |
| | | | ¼ lower part | <0.001* | 0.042 | 0.085 |
| Linear (millimetre) | 1/3 exposed | No deficiency | ¼ lower part | 0.862 | -0.027 | 0.017 |
| | | | ¼ upper and lower part | 0.008* | -0.051 | -0.006 |
| | | ¼ lower part | No deficiency | 0.862 | -0.017 | 0.027 |
| | | | ¼ upper and lower part | 0.032* | -0.046 | -0.002 |
| | | ¼ upper and lower part | No deficiency | 0.008* | 0.006 | 0.051 |
| | | | ¼ lower part | 0.032* | 0.002 | 0.046 |
| | 2/3 exposed | No deficiency | ¼ lower part | 0.884 | -0.003 | 0.005 |
| | | | ¼ upper and lower part | 0.150 | -0.007 | 0.001 |
| | | ¼ lower part | No deficiency | 0.884 | -0.005 | 0.003 |
| | | | ¼ upper and lower part | 0.055 | -0.008 | 0.000 |
| | | ¼ upper and lower part | No deficiency | 0.150 | -0.001 | 0.007 |
| | | | ¼ lower part | 0.055 | 0.000 | 0.008 |
| | full exposed | No deficiency | ¼ lower part | 0.813 | -0.002 | 0.004 |
| | | | ¼ upper and lower part | 0.064 | -0.006 | 0.000 |
| | | ¼ lower part | No deficiency | 0.813 | -0.004 | 0.002 |
| | | | ¼ upper and lower part | 0.014* | -0.007 | -0.001 |
| | | ¼ upper and lower part | No deficiency | 0.064 | 0.000 | 0.006 |
| | | | ¼ lower part | 0.014* | 0.001 | 0.007 |

significance, it is important to note its proximity to the threshold. This suggests that a larger sample size or alternative statistical approaches may yield different outcomes. Additionally, the study revealed that when the scan body was exposed only 1/3 of its length and a 1-point scan body alignment method was used, it led to the highest degree of scan deficiency and the greatest deviations in implant position in both angular and linear dimensions. This implies that incomplete acquisition of the scan body and limited scan body exposure may hinder the CAD software's ability to accurately recognize and position the implant replica. These findings align with a previous study [13], which utilized a different scanning system and also

demonstrated the adverse impact of inadequacies in scanned images of a scan body on implant positioning accuracy in CAD software. Similar to our study, the previous research identified increasing deviations with higher deficiency levels. The extent of the deviations in the previous study, reaching up to 0.081.6 ± 0.002 mm in linear discrepancy and 0.26 ± 0.01 degrees in angular discrepancy at 15% of scan body deficiency. While in our investigation, we introduced an additional variable of implant depth, and the highest deviations at 0.084 ± 0.068 mm and 0.237 ± 0.059 degrees, were observed when the implant was placed at a greater depth, only 1/3 of scan body exposed, and with a scan deficiency in the 1/4 upper and lower part of the scan body. These congruent results underscore the critical importance of addressing scan body image deficiencies to ensure precise virtual implant positioning. The fact that both studies, despite using different scanning systems and slightly different experimental setups, produced comparable results regarding the impact of scan deficiency on implant accuracy further validates the significance of these observations.

Accurate scan body alignment in CAD software relies on the proper identification and capture of the scan body's flat surface. This step is essential for ensuring the scan body is correctly positioned in the virtual setting, which in turn affects the accuracy of virtual implant positions. In the present study, despite establishing a complete image of the scan body's flat surface in all scanned data, it was observed that a 1-point alignment method exhibited higher susceptibility to error compared to a 3-point alignment method.

The results therefore emphasize the significance of accurate scan body alignment techniques in CAD software for achieving precise virtual implant positions. Previous studies [4, 17, 18] that solely assessed scan body position deviations might not provide a comprehensive evaluation of overall accuracy in virtual implant placement. It is crucial to consider accurate digitization of intraoral scans and capturing sufficient scanned objects for obtaining precise virtual implant positions. However, the results of this study suggest that accurate scan body alignment in CAD software plays a critical role in ensuring overall accuracy in virtual implant positions. These findings underscore the importance of precise scan body alignment, particularly when the scan body is exposed only 1/3 of the total length. Employing a 3-point alignment approach can enhance accuracy in virtual implant placement, especially in cases where the implant is positioned deeper. Conversely, when the scan body is exposed at least two-thirds of the total length, the impact of scan body alignment on virtual implant placement accuracy is minimal. Future research should focus on evaluating and comparing different scan body alignment techniques in commercial dental CAD software to determine the most effective approach. Enhancing accuracy and reliability in virtual implant placement can ultimately improve clinical outcomes for patients.

Assessing the accuracy of virtual implants in varieties of clinical situations may allow a true demonstration of the effectiveness of intraoral scanners. It is crucial to avoid any alterations or modifications to the scan body, as these may interfere with the software's ability to accurately recognize the scan body position [19]. In cases where implants are misaligned or located in areas with limited access, a smaller scanning tip for the intraoral scanner can be beneficial to ensure complete image acquisition, especially when implants are placed at greater depths. However, it is important to acknowledge that there may be instances where a digital workflow is not feasible, and a conventional impression and laboratory scanner may be necessary. These considerations highlight the practical implications and limitations of using intraoral scanners in implant dentistry.

Despite the valuable findings, there are certain limitations that should be acknowledged. Firstly, the study utilized simulated implant models, which may not fully replicate the complexity and variability of clinical scenarios. As mentioned earlier, further research using actual patient cases is needed to validate these results. Secondly, the study focused on specific CAD

software and scan body alignment techniques. The findings may not be directly applicable to other software or alignment methods, and therefore, caution should be exercised when generalizing the results to different systems. Additionally, the study evaluated the accuracy of virtual implant positions but did not assess the clinical outcomes or long-term success of the restorations based on these positions. Further investigations involving clinical studies are required to determine the impact of scan body image capture deficiency and alignment methods on the clinical efficiency as well as overall short- and long-term treatment outcomes. Lastly, the sample size in this study was relatively small, which may limit the generalizability of the results. A larger sample size would provide more robust statistical analyses and a better understanding of the relationship between scan body deficiencies, implant depth, and virtual implant accuracy.

## Conclusion

In conclusion, this study highlights the importance of accurate scan body image capture and alignment techniques in achieving precise virtual implant positions. Deficiencies in scan body acquisition, particularly in deep soft tissue situations, can lead to deviations in both angular and linear positioning of virtual implants. Employing appropriate scan body alignment methods such as a 3-point alignment approach demonstrates better accuracy compared to a 1-point alignment. Modifying scan bodies should be done with cautions since it can interfere with their recognition by CAD software. Complex clinical scenarios require the need for further research on different software and alignment methods. Although limitations exist, including the use of simulated implant models and a relatively small sample size, this study lays the groundwork for optimizing digital implant workflows to improve treatment outcomes in implant dentistry.

## Supporting information

**S1 File.**
(DOCX)

**S1 Data.**
(XLSX)

## Acknowledgments

The authors would like to express their sincere gratitude to the research team, staff, and Faculty of Dentistry of Mahidol University for their invaluable support and provision of exceptional facilities that enabled the successful completion of this manuscript. Their dedication, expertise, and hard work have been instrumental in advancing the field of dentistry and improving oral health outcomes.

## Author Contributions

**Conceptualization:** Pobploy Petchmedyai, Prakan Thanasrisuebwong.

**Data curation:** Pobploy Petchmedyai.

**Formal analysis:** Pobploy Petchmedyai.

**Investigation:** Pobploy Petchmedyai, Prakan Thanasrisuebwong.

**Methodology:** Pobploy Petchmedyai, Prakan Thanasrisuebwong.

**Project administration:** Prakan Thanasrisuebwong.

**Resources:** Pobploy Petchmedyai, Prakan Thanasrisuebwong.

**Software:** Pobploy Petchmedyai.

**Supervision:** Prakan Thanasrisuebwong.

**Validation:** Pobploy Petchmedyai, Prakan Thanasrisuebwong.

**Visualization:** Pobploy Petchmedyai, Prakan Thanasrisuebwong.

**Writing – original draft:** Pobploy Petchmedyai.

**Writing – review & editing:** Pobploy Petchmedyai, Prakan Thanasrisuebwong.

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
