## [Decision Letter · Decision Letter 0]

23 May 2023

PONE-D-23-11871Optimizing Digital Implant Impressions: The Importance of Scan Body Image Deficiency and Alignment in Varying Scan Body ExposuresPLOS ONE

Dear Dr. Thanasrisuebwong,

Thank you for submitting your manuscript to PLOS ONE. After careful consideration, we feel that it has merit but does not fully meet PLOS ONE’s publication criteria as it currently stands. Therefore, we invite you to submit a revised version of the manuscript that addresses the points raised during the review process.

We look forward to receiving your revised manuscript.

Kind regards,

Luca Bertolaccini, M.D., Ph.D.

Academic Editor

PLOS ONE

Journal Requirements:

Additional Editor Comments:

The authors are thanked for this submission.

The reviewers have emphasised issues that require a careful and thorough manuscript revision.

No commitment to publication can be made at this point.

Reviewers' comments:

Reviewer's Responses to Questions

**Comments to the Author**

1. Is the manuscript technically sound, and do the data support the conclusions?

Reviewer #1: Yes

Reviewer #2: Yes

2. Has the statistical analysis been performed appropriately and rigorously? 

Reviewer #1: Yes

Reviewer #2: Yes

3. Have the authors made all data underlying the findings in their manuscript fully available?

Reviewer #1: Yes

Reviewer #2: Yes

4. Is the manuscript presented in an intelligible fashion and written in standard English?

Reviewer #1: Yes

Reviewer #2: Yes

5. Review Comments to the Author

Reviewer #1: This article aimed to investigate the influences of scan body’ deficiency and alignment on intraoral scanners’ accuracy in the field of implant dentistry.

My comments are as follows:

1. It’s suggested that a native English speaker embellish this manuscript.

2. It’s suggested to add more explanations to some terminology, e.g., scan body and deficiency,

and add some explanations on figures especially Fig. 3.

3. What is the significance of depth control? As stated above, the depth is influenced by the patient's real situation, and it is not artificially possible to increase or decrease the depth intentionally to improve the accuracy. It will be better if the clinical significance can be explained clearly.

4. How did the authors control the consistency of the deficiency?

5. How are the cylinders generated? Does it refer to the scan body or the geometric cylinder generated from the scan body?

6. It’s suggested to do more statistical analysis of the data in this research. And more explanations can be presented in result section.

7. It’s suggested to represent more information of this model, especially the length of the scan body.

8. It’s suggested to label for the structures in Fig. 5.

9. It’s suggested to provided figures with higher resolution.

I feel that your manuscript does not reach a particular level of technical or scientific advancement and may not be appropriate for this journal's readership to allow for further consideration.

Thank you.

Reviewer #2: Abstract

-It needs to rewrite the whole abstract. Most of the abstract about introdctuion and nothing about the actual study. For example “This study provides valuable insights into the factors that could affect the accuracy of virtual implant positions obtained from digital impressions. “How ? nothing mentioned.

-There is no 0.000 P value. Change it to <0.001

-Who did the scan ? all of them by the same person ? did he/she has enough experience to use it ? who did the defect ? ….etc

-P value should capital and italic

-What are the limitations of this study ?

6. PLOS authors have the option to publish the peer review history of their article (what does this mean?). If published, this will include your full peer review and any attached files.

Reviewer #1: No

Reviewer #2: **Yes: **Zuhair Natto

---

## [Author Response · Author response to Decision Letter 0]

6 Jul 2023

Response to reviewer 1 comments: 

 We would like to express our sincere gratitude for your valuable feedback on our manuscript. We appreciate the time and effort invested in conducting a comprehensive evaluation of our work. Your comments and suggestions have been immensely helpful in identifying areas that required revision and improvement. We also appreciate your feedback and concern regarding the level of technical or scientific advancement in our manuscript. We acknowledge that our research may not introduce groundbreaking advancements in the field; however, we believe that our study provides valuable insights and contributes to the existing knowledge base in digital implant prostheses workflows.

 Our research focuses on simulating scan body deficiencies and investigating the implications of incomplete scan body capture. Using advanced CAD software, we have successfully recreated various clinical scenarios that replicate real-life situations where incomplete scans can occur. These scenarios encompass factors such as malposed adjacent teeth, extrusion of opposing teeth, implant positioning challenges, and limited mouth opening, particularly in the scanning area of the second molar. These examples highlight the potential limitations in scanning access, which can have a direct impact on the overall capture of the scan body. In turn, this can significantly affect the accuracy of digital implant impressions. Moreover, the significance of depth control in our study lies in its representation of a variable that can present challenges in clinical practice. In certain scenarios, such as in the anterior esthetic zone or areas with limited bone availability, the placement of implants at greater depths may be required, or even the length of implant scan body of some implants systems are too short. This can result in a decrease in the required scan body exposure for accurate image capture using intraoral scanners. By meticulously replicating these scenarios, we aim to gain a comprehensive understanding of the implications associated with incomplete scan body capture.

 Our research is of significant importance as it provides valuable insights for professionals in the fields of digital dentistry, implantology, and prosthodontics. The outcomes of our study have the potential to advance digital implant workflows and assist in the development of strategies aimed at overcoming the challenges associated with incomplete scan body capture. For instance, our research suggests potential methods to enhance the accuracy of impressions during the CAD software-based prosthesis design phase. These methods may include the utilization of 3-point alignment techniques or considering the use of conventional impressions as an alternative to intraoral scanners. By exploring these approaches, we aim to offer practical solutions that can mitigate the impact of incomplete scan body capture on the overall treatment process. Our findings can help clinicians improve their prosthesis design workflows and optimize the accuracy of digital implant impressions. These insights may prove valuable in enhancing patient outcomes and advancing the field of digital dentistry as a whole. 

We have carefully reviewed and considered all of the comments and have made significant revisions to the manuscript accordingly. Each of your points was addressed in the following sections and separated in 3 sections: Comment, Response, Text change. 

Changes made in the manuscript are highlighted in yellow, for your convenience, so they can be easily identified. It's worth noting that beyond the revisions you suggested, we have also performed further edits that we believed necessary to improve the overall quality and clarity of the manuscript. These changes, performed for the enhancement of our manuscript, were tracked in red in the revised with track change manuscript file. We respectfully request to reconsider your assessment and take into account the contribution our study makes to the field. We are open to any more suggestions or recommendations for further improvement that would enhance the quality and relevance of our manuscript. Please find all response in the attached file " Response to reviewer 1 comment". 

Response to reviewer 2 comments: 

We would like to express our gratitude for your valuable feedback on our manuscript. We sincerely appreciate their time and effort in providing a thorough evaluation. Your comments have helped us identify the areas that require revision and improvement. In the following sections, we address each of your points separated in 3 sections: Comment, Response, Text change. Highlight the revisions made in grey color for your convenience. It's worth noting that beyond the revisions you suggested, we have also performed further edits that we believed necessary to improve the overall quality and clarity of the manuscript. These changes, performed for the enhancement of our manuscript, were tracked in red in the revised with track change manuscript file. Please find all response in the attached file " Response to reviewer 2 comment".

---

## [Decision Letter · Decision Letter 1]

24 Jul 2023

PONE-D-23-11871R1Optimizing Digital Implant Impressions: Evaluating the Significance of Scan Body Image Deficiency and Alignment under Varied Scan Body ExposuresPLOS ONE

Dear Dr. Thanasrisuebwong,

Thank you for submitting your manuscript to PLOS ONE. After careful consideration, we feel that it has merit but does not fully meet PLOS ONE’s publication criteria as it currently stands. Therefore, we invite you to submit a revised version of the manuscript that addresses the points raised during the review process.

We look forward to receiving your revised manuscript.

Kind regards,

Luca Bertolaccini, M.D., Ph.D.

Academic Editor

PLOS ONE

Journal Requirements:

Additional Editor Comments :

I would urge the authors to carefully read the remarks of Reviewers to consider these in the revised paper.

Reviewers' comments:

Reviewer's Responses to Questions

**Comments to the Author**

1. If the authors have adequately addressed your comments raised in a previous round of review and you feel that this manuscript is now acceptable for publication, you may indicate that here to bypass the “Comments to the Author” section, enter your conflict of interest statement in the “Confidential to Editor” section, and submit your "Accept" recommendation.

Reviewer #3: All comments have been addressed

Reviewer #4: All comments have been addressed

2. Is the manuscript technically sound, and do the data support the conclusions?

Reviewer #3: Yes

Reviewer #4: Yes

3. Has the statistical analysis been performed appropriately and rigorously? 

Reviewer #3: Yes

Reviewer #4: Yes

4. Have the authors made all data underlying the findings in their manuscript fully available?

Reviewer #3: Yes

Reviewer #4: Yes

5. Is the manuscript presented in an intelligible fashion and written in standard English?

Reviewer #3: Yes

Reviewer #4: Yes

6. Review Comments to the Author

Reviewer #3: Dear The Authors,

The Authors have addressed all the comments. I appreciate this.

Regards,

Reviewer

Reviewer #4: 1.Can this article provide a comparison with other similar research findings, if possible.

2.The clarity of the images is insufficient and needs to be modified according to the format required by the journal.

7. PLOS authors have the option to publish the peer review history of their article (what does this mean?). If published, this will include your full peer review and any attached files.

Reviewer #3: No

Reviewer #4: No

---

## [Author Response · Author response to Decision Letter 1]

5 Sep 2023

Response to reviewer 3 : 

I am writing to express my sincere gratitude for your time and efforts in reviewing our manuscript titled " Optimizing Digital Implant Impressions: Evaluating the Significance of Scan Body Image Deficiency and Alignment under Varied Scan Body Exposures." Your positive feedback is truly encouraging, and it motivates us to continue improve our work. We appreciate your efforts made to address the specific aspects highlighted during the review process. Your thorough assessment on our manuscript, making the final version more robust and informative. We are thankful for the opportunity to have our work reviewed by someone as experienced and knowledgeable as yourself. Please know that we are open to any further suggestions or insights you may have, and we welcome any additional feedback that could further enhance the impact of our research. Once again, thank you for being a part of the peer review process and for your valuable assistance. 

Response to reviewer 4 : 

Thank you for your valuable feedback regarding additional comparison with other similar research findings and the clarity of the images in our manuscript. We sincerely appreciate your comments, and we have taken them into consideration during the revision process.

 During the literature review process, we found that only one study had specifically assessed the impact of scan body deficiency on the accuracy of virtual implant positioning in CAD software. This study, conducted by Park in 2020, utilized a different scanning system and provided valuable insights into the adverse consequences of inadequacies in scanned images of a scan body on accuracy of virtual implant position. The scarcity of research focusing on this particular aspect highlights the novelty and significance of our study in contributing additional evidence and further understanding the critical importance of addressing scan body image deficiencies for precise virtual implant position. 

A comparison with this research has already been included in the Discussion section of our manuscript as below.

 “These findings are in agreement with a previous study [13], which utilized a different scanning system and also demonstrated the adverse impact of inadequacies in scanned images of a scan body on implant positioning accuracy in CAD software. Similar to our study, the previous research identified increasing deviations with higher deficiency levels. The extent of the deviations in the previous study, reaching up to 0.081.6 ± 0.002 mm in linear discrepancy and 0.26 ± 0.01 degrees in angular discrepancy at 15% of scan body deficiency. While in our investigation, we introduced an additional variable of implant depth, and the highest deviations at 0.084 ± 0.068 mm and 0.237 ± 0.059 degrees, were observed when the implant was placed at a greater depth, only 1/3 of scan body exposed, and with a scan deficiency in the 1/4 upper and lower part of the scan body. These congruent results underscore the critical importance of addressing scan body image deficiencies to ensure precise virtual implant positioning. The fact that both studies, despite using different scanning systems and slightly different experimental setups, produced comparable results regarding the impact of scan deficiency on implant accuracy further validates the significance of these observations.”

 Regarding the clarity of the images, we are pleased to inform you that we have made significant improvements to all the figures in accordance with the journal's requirements. To achieve this, we utilized the Preflight Analysis and Conversion Engine (PACE) digital diagnostic tool, as suggested by the journal. This tool ensured that all our figure files now meet the specified guidelines. We followed the provided instructions on the PACE platform, which allowed us to make the necessary adjustments and modifications to each figure file.

 All figure files are now in TIFF and the dimensions of the figures are within the specified range, with a width between 789 and 2250 pixels (at 300 dpi) and a maximum height of 2625 pixels (at 300 dpi). The resolution of each figure is now set at 300 to 600 dpi, and we have successfully reduced the file sizes to be under 10 MB. as per the journal's instructions. We believe that these modifications have significantly enhanced the clarity and quality of the figures in our manuscript, making them more suitable for publication. 

Once again, we express our gratitude for your thorough evaluation and constructive feedback, which has undoubtedly improved the quality of our work. We look forward to any further comments or suggestions you may have during the review process.

---

## [Editor Report · Decision Letter 2]

6 Sep 2023

Optimizing Digital Implant Impressions: Evaluating the Significance of Scan Body Image Deficiency and Alignment under Varied Scan Body Exposures

PONE-D-23-11871R2

Dear Dr. Thanasrisuebwong,

We’re pleased to inform you that your manuscript has been judged scientifically suitable for publication and will be formally accepted for publication once it meets all outstanding technical requirements.

Kind regards,

Luca Bertolaccini, M.D., Ph.D.

Academic Editor

PLOS ONE

Additional Editor Comments (optional):

The manuscript is acceptable for publication.
---

## [Editor Report · Acceptance letter]

12 Sep 2023

PONE-D-23-11871R2 

Optimizing Digital Implant Impressions: Evaluating the Significance of Scan Body Image Deficiency and Alignment under Varied Scan Body Exposures 

Dear Dr. Thanasrisuebwong:

I'm pleased to inform you that your manuscript has been deemed suitable for publication in PLOS ONE. Congratulations! Your manuscript is now with our production department. 

Kind regards, 

on behalf of

Dr. Luca Bertolaccini 

Academic Editor

PLOS ONE